# Classification of Flying Drones Using Millimeter-Wave Radar: Comparative Analysis of Algorithms Under Noisy Conditions

**DOI:** 10.3390/s25030721

**Published:** 2025-01-24

**Authors:** Mauro Larrat, Claudomiro Sales

**Affiliations:** Instituto de Ciências Exatas e Naturais, Universidade Federal do Pará, Rua Augusto Corrêa, 01 Guamá, CEP, Belém 66075-110, PA, Brazil; cssj@ufpa.br

**Keywords:** drone detection, millimeter-wave radar sensor, machine learning

## Abstract

This study evaluates different machine learning algorithms in detecting and identifying drones using radar data from a 60 GHz millimeter-wave sensor. These signals were collected from a bionic bird and two drones, namely DJI Mavic and DJI Phantom 3 Pro, which were represented in complex form to preserve amplitude and phase information. The first benchmarks used four algorithms, namely long short-term memory (LSTM), gated recurrent unit (GRU), one-dimensional convolutional neural network (Conv1D), and Transformer, and they were benchmarked for robustness under noisy conditions, including artificial noise types like white noise, Pareto noise, impulsive noise, and multipath interference. As expected, Transformer outperformed other algorithms in terms of accuracy, even on noisy data; however, in certain noise contexts, particularly Pareto noise, it showed weaknesses. For this purpose, we propose Multimodal Transformer, which incorporates more statistical features—skewness and kurtosis—in addition to amplitude and phase data. This resulted in a improvement in detection accuracy, even under difficult noise conditions. Our results demonstrate the importance of noise in processing radar signals and the benefits afforded by a multimodal presentation of data in detecting unmanned aerial vehicle and birds. This study sets up a benchmark for state-of-the-art machine learning methodologies for radar-based detection systems, providing valuable insight into methods of increasing the robustness of algorithms to environmental noise.

## 1. Introduction

Drones or unmanned aerial vehicles (UAVs) are employed in various sectors these days. Although there are many advantages to UAV technology, the quick growth has also posed important issues regarding privacy, safety, and security. These concerns have fueled the growth of the counter-UAV (C-UAV) market, ranging from military suppliers to innovative startups, and they are focused on detecting, tracking, identifying, and neutralizing unauthorized drones [1,2]. The detection of UAVs is particularly pronounced in environments where traditional sensors face limitations, such as low radar cross-section areas or during complex flight stages [3,4,5].

Several UAV detection methods have been explored in the literature, including acoustic monitoring [6,7,8,9,10,11,12], visible light [13,14,15,16,17,18], infrared [19,20], radar detection [21,22,23,24,25,26,27,28,29,30,31], and radio spectrum monitoring [32,33,34,35,36]. However, each method has its limitations. Acoustic monitoring is susceptible to ambient noise, and visible light detection can be impaired by weather conditions or obstacles such as buildings. As UAVs become smaller and less conspicuous, both infrared and radar technologies cannot maintain their effectiveness.

Among these methods, radar detection, particularly millimeter wave (mmWave), stands out due to its resilience to weather conditions and ability to provide high-resolution detection. Unlike acoustic methods, which are influenced by environmental noise, or visible light detection, which requires a clear line of sight, radar is robust in diverse operational contexts [23,28]. Additionally, mmWave radar sensors, operating in the 24 GHz to 300 GHz range can effectively detect small objects with fine precision, making them suitable for UAV detection from both recreational and security perspectives [23,29].

Currently, multi-target active domain adaptation (MT-ADA) has been set up for active sample selection and simultaneous domain alignment. This advancement in domain adaptation strategies has significantly improved radar detection capabilities, though primarily in image classification [37]. Recent innovations have demonstrated the effectiveness of unsupervised domain adaptation in addressing domain shifts in synthetic aperture radar for target detection. Shi et al.’s [38] study proposed a novel UDA framework that aligns domain-level and class-level features to facilitate SAR target classification from simulated to real-world data. Chen and Jia [39] likewise posited a UDA method for bringing-through-wall radar imaging. This approach improved detection performance under limited labeled data. In addition, since the creation of synthetic radar detection (SAR), Guo et al. [40] presented a domain-adaptive faster R-CNN framework, which is effective in overcoming these domain disparities, even with small training datasets. These studies reiterate the importance of domain adaptation in improving radar-based detection systems in diverse environments of operation. Nevertheless, the key principles of active domain adaptation could easily be implemented for radar use, particularly in scenarios characterized by multiple targets and varying conditions.

The application of multi-kernel-size feature fusion in the convolutional neural network (MKSFF CNN) model has been applied for the target classification of SAR, achieving greater completeness in representation for more useful classification [41]. In recent years, much advancement has been made in multi-scale feature fusion and deep learning architecture to detect and classify UAVs. For example, one study presented a multi-scale feature fusion algorithm known as DA-FS SSD, which is enhanced by incorporating dilated convolution and attention mechanisms into the image analysis of the object in UAV images [42]. Another introduced a multi-scale aerial image detection method for UAVs based on adaptive feature fusion to address the difficulty of small target detection within cluttered backgrounds [43]. Researchers have developed a lightweight CNN model applying depthwise separable convolutions to classify UAV images with high accuracy and low computational complexity [44].

Moreover, state-of-the-art results were obtained when the convolutional neural networks were fed into long short-term memory networks in applications related to UAVs. A hybrid CNN-LSTM framework was developed for automatically detecting anomalies in farmland by the aerial images collected from UAVs, capturing spatial and temporal characteristics for further classification enhancement [45]. A multi-label classification of UAV sounds was also proposed, incorporating the use of a stacked bidirectional LSTM-CNN architecture to improve the ability of the model to simultaneously classify many UAV sounds [46]. The accuracy in identifying and classifying UAVs using millimeter-wave radar data would improve through such feature-fusion architectures.

This study discusses the 60 GHz millimeter-wave radar data collected by a bionic bird and two types of drones, DJI Mavic and DJI Phantom 3 Pro (see the illustration of these classes in Figure 1), and they are then applied to various machine learning algorithms. The performances of several models such as LSTM, GRU, CNN, and Transformer are compared, considering their robustness when noise is artificially added to the dataset. Additionally, the Multimodal Transformer architecture is proposed, leveraging amplitude, phase, skewness, and kurtosis features for improved detection. These findings aim to enhance detection accuracy in noisy environments and provide a benchmark for advanced radar-based UAV detection techniques. These results hold significant implications for public safety, logistics, and agriculture, where identifying small drones is increasingly critical for security and operational efficiency.

## 2. Overview of the Related Research

Recently, the detection and analysis of micro-Doppler signatures from UAVs, particularly multi-rotor drones, has received considerable attention. Yang et al. (2022) [21] proposed a method to improve the rotor blade micro-Doppler parameters estimation through complex variational mode decomposition (CVMD) and singular value decomposition (SVD). Meanwhile, this method addresses some of the limitations of traditional methods.

Hanif et al. (2022) [22] presented a comprehensive review of radar micro-Doppler signature analysis techniques, with a particular focus on target classification. Their work highlights the evolution of the feature extraction methods and limitations of existing approaches. This provides a solid background for addressing small target detection—which aligns with the challenges of classifying these drones or aerial targets.

Soumya et al. (2023) discussed recent advances in radar sensor technology by focusing on mmWave radars [23]. Their review highlights the possible combination of radar data with machine learning algorithms for an improved overall performance in different applications including UAV detection. This inclusion of machine learning fits into our research approach to improve the way IRIS UAV detections can be completed with neural network models.

Kumawat et al. (2022) proposed a deep convolutional neural network (DCNN) for detection and classification of SUAVs via their micro-Doppler signatures [24]. Moreover, their dataset “DIAT-*μ*SAT” provides full radar signature information, which can further enhance the accuracy of detection. These efforts in under-the-hood DCNNs and custom datasets are close to our heart as we refine methods for UAV detection using radar data and emerging machine learning approaches.

Lu et al. (2024) [27] illustrated further advances in the detection using radar, where the use of exponential complex sinusoidal basis (ECSB) allowed for the extraction of micro-Doppler signatures with a high classification accuracy. Similarly, Kang et al. (2024) [28] provided valuable experimental data using 60 GHz mmWave for both drone and bionic bird micro-Doppler analysis by placing further emphasis on how accurate feature extraction is important in UAV detection.

A related study by Narayanan et al. (2023) [29] utilized TensorFlow to classify radar-based micro-Doppler spectrograms. The model resulted in extremely high accuracy when tasked with differentiating between the drone types/birds. Their work thus further underlines the fact that the choice of robust classification models is an important consideration and constitutes one cornerstone of our approach for the detection of UAVs.

Finally, Fan et al. (2023) [31] proposed a clutter suppression method using an orthogonal matching pursuit, which effectively enhances the clarity of the micro-Doppler signal from environmental interference. This technique is particularly relevant for improving the performance of detection under poor environmental conditions.

## 3. Foundational Theories and Methodological Approaches

### 3.1. The Detected Signal Representation

The radar signal model of multirotor UAVs is intrinsically based upon the interaction of radar with both rotor blades and the body of the drone. In this model, the reflected radar signal takes a specific form in micro-Doppler signatures. Indeed, those signatures are produced not only by the UAV itself, but also by the action of radar on the flapping wings and bodies of birds. Mathematically, the micro-Doppler signal can be expressed as follows [27,31]:(1)$$s\left(t\right)=L\exp\left(-j\frac{4\pi }{\lambda }R_{0}\right)\sum_{p=0}^{P-1}\sum_{\kappa =0}^{K-1}\mathrm{sinc}\left(\psi_{p,\kappa }\left(t\right)\right)\exp\left(-j\psi_{p,\kappa }\left(t\right)\right),$$
where:*L* is a constant related to the radar system;λ is the wavelength of the transmitted radar signal;$R_{0}$ is the distance between the radar and the center of the UAV;*P* is the number of rotors on the UAV;*K* is the number of blades per rotor;$\psi_{p,\kappa }\left(t\right)$ is the Doppler phase associated with blade κ of rotor *p*;$\mathrm{sinc}(\psi_{p,\kappa }\left(t\right))$ models the spatial response of the Doppler shift of each blade.

The formula represents the signal captured by the mmWave radar sensor datasets [25,28], and it also considers that it has been fitted to the specific context of drone detection or similar objects, including the micro-Doppler behavior in the case of a captured signal. The latter is a complex number array, which may be considered typical for the signals recorded by mmWave radar. These complex numbers encode the magnitude and phase of the reflected wave, both of which are fundamental to micro-Doppler calculations.

The key ingredients in the mathematical modeling of micro-Doppler behavior include the following: *L*, which is a constant parameter associated with the radar system-the value that should be adjusted to conform to specific conditions of the radar setup; $R_{0}$, which denotes the distance between the radar and object-under-test—a drone in our case—and it determines the amplitude and phase of the complex data acquired. The other parameters are as follows: *P* denotes the number of rotors, while *K* denotes the number of blades per rotor; these two jointly determine the frequency modulation pattern observed in the radar signal.

The $\exp\left(-j\frac{4\pi }{\lambda }R_{0}\right)$-term is the interaction with the radar, which actually is the phase of the electromagnetic wave arriving back to the radar modulated by the distance from the radar to the object. It reflects the phase changes observed in the complex data. $\mathrm{sinc}\left(\psi_{p,\kappa }\left(t\right)\right)\exp\left(-j\psi_{p,\kappa }\left(t\right)\right)$ models phase modulation due to the movement of the rotor blades and gives the micro-Doppler signature that is typical for multirotor UAVs. This can also be interpreted as phase variation in the captured complex signal, whereby one notices changes continuously in both the real and imaginary parts with respect to time.

Although this is a formula that mathematically models an ideal scenario, it captures all the main features in the behavior implicit in the radar data. However, the actual signal will be noisy, with fluctuations in environmental influence, several reflections, and disturbances around the sensor. In that aspect, filtering or any other noise separation techniques may be necessary to further refine the model and fully apply to the context at hand.

### 3.2. Noise Models in the Original Signal Representation

Accurately introducing different types of noise to the modeling of radar signals is crucial for enhancing both the reliability and precision of the model. Realistic radar signals are susceptible to different forms of noise; each has a different definition and depth of influence on the integrity of the signal.

White noise is typically modeled as Gaussian noise, and it represents the electronic noise intrinsic to radar systems. It has a zero mean and a variance $\sigma^{2}$, which is mathematically described as(2)$$n_{w}\left(t\right)∼N\left(0,\sigma_{w}^{2}\right).$$

It reduces signal clarity, especially in areas with significant electronic interference.

Impulsive noise: sporadic high-amplitude pulses that are modeled by a Poisson distribution with rate λ: (3)$$n_{i}\left(t\right)=\mathrm{Poisson}\left(\lambda \right).$$

This type of noise may result from sudden interference, an electrical discharge, or an abrupt change in the source signal that masks and/or distorts the signal being transmitted by the radar.

Noise following the Pareto distribution can be modeled as follows: (4)$$f_{p}\left(x;\alpha ,x_{m}\right)=\frac{\alpha x_{m}^{\alpha }}{x^{\alpha +1}},$$
where α and $x_{m}$ are parameters of the distribution. This noise has heavy tails. It can bring large, infrequent disturbances in the signal as representation of a rare event can significantly impact radar data interpretation.

Multi-path interference: The reflection of radar signals from various surfaces to the receiver causes interference among several late-time copies of the same signal. This scenario arises when surroundings contain high-rise buildings, or if the terrain is complex enough to significantly alter the waveform of the received signal. The resultant interference may degrade target detection or tracking due to error. Multipath noise can be modeled mathematically as follows:(5)$$n_{m}\left(t\right)=\sum_{k=1}^{K}\alpha_{k}s\left(t-\tau_{k}\right),$$
where $\alpha_{k}$ represents the magnitude of the kth multi-path component, s(t) is the transmitted radar signal, and $\tau_{k}$ is the time delay associated with each multi-path reflection. With different noise types considered, the extended representation of the radar signal is given as follows:(6)$$s\left(t\right)=L\exp\left(-j\frac{4\pi }{\lambda }R_{0}\right)\sum_{p=0}^{P-1}\sum_{\kappa =0}^{K-1}\mathrm{sinc}\left(\psi_{p,\kappa }\left(t\right)\right)\exp\left(-j\psi_{p,\kappa }\left(t\right)\right)+n_{w}\left(t\right)+n_{i}\left(t\right)+n_{p}\left(t\right).$$

This general model provides better processing and analysis of the signals; hence, an appropriate method for detecting and interpreting radar data with many sources of noise and effects of interference. This model, since there will only be one term to account for each type of noise, can accommodate myriad disturbances associated with radar signals. Finally, it offers a close approximation to actualities in the operation of radar, hence offering robustness and more realistic accuracy in the representation of radar signals.

### 3.3. Machine Learning Algorithms for Drone Detection

Recent advances in signal processing and machine learning have considerably enhanced the performance of sensors in detecting UAVs [3,4,5]. Higher detection accuracy and reliability have, therefore, enabled the identification of small-sized drones that become relevant in many applications related to surveillance and recreational use, such as, for instance, [6,8,10,13,14,15,16,17,24,27,29,32,33,34,35]. These, when integrated with radar systems, ensure superior spatial resolution and, more importantly, effective operations in difficult environmental conditions for more robust UAS detection solutions [21,22,23,24,27,29].

Several machine learning models were implemented to investigate their performances related to the detection and classification of UAVs. In particular, both LSTM networks [5,8,11,18] and GRU [11,18] have the particular advantageous capability of temporal dependency and sequential data handling. This is crucial, as radar signals are time-series data, and flying patterns of aerial objects are not steady in certain conditions. They are also particularly superior in representing the spatial hierarchies and local patterns that can be found in radar data. Therefore, it can effectively capture distinguishable features in decomposed signal components.

CNNs are adept at catching such spatial hierarchies and local patterns in radar data, serving the purpose intrinsically for distinct feature identification from decomposed signal components [4,5,8,10,13,14,15,16,17,22,24,34,35]. Furthermore, Transformer models are also employed for their ability to handle long-range dependencies and self-attention mechanisms with improved efficiency [15,16]. These are highly crucial to understand the complex interaction within the radar signals themselves for correct classification. Among the selected algorithmic approaches, some bear respective strengths to address the nature of the data with a particular interest in sequential and spatial features to enhance overall performance and reliability for any given real-world scenario of detection and classification.

Many drone detection algorithms are based on machine learning methods applied either in conjunction with or as standalone alternatives to signal decomposition techniques [7,8,9,21,22,27,33] and attention mechanisms [13,15,16]. Other methods rely on visualization techniques of the signals based on spectrograms [6,22,29,36] to detect and classify drone activities. Other approaches are simply statistical techniques [12,18,30,36], and signal analysis is performed without machine learning.

While traditional methods are extremely efficient in analyzing patterns or trends within data, machine learning algorithms perform extremely well when modeling complex, nonlinear relationships that may not be as saliently evident by purely statistical techniques. This algorithm can learn features automatically from enormous datasets, reducing the need for manual feature extraction.

Machine learning models can adapt and improve over time; thus, their functionality will be even better in environments that have considerable variability and challenge—even highly interfering or noisy ones. Unlike visual signal methods, which may depend on specific patterns or thresholds, machine learning can generalize over different signal types and conditions, hence making the robustness and accuracy of drone detection better.

### 3.4. Overview of the Dataset Employed in This Research

Kang et al. (2024) [28] presents the radar signal collection collected through a 60 GHz radar sensor. This data structure falls within three dimensions, which are important for the functionality of the sensor and its interaction with the environment.

The first dimension stretches across 1000 entries, which represent distinct time instances or “frames” in temporal sequence that capture the motion or otherwise appearance of objects (drone, bird, etc.) present in the radar field of view. At an extremely high level, these frames represent the snapshots of scenes observed during one period of running.

The second part has 128 subdivisions in each frame; those windows act as smaller time windows giving enhanced temporal resolution. This allows sensing the micro-Doppler phenomenon of rapid wing flapping or rotor motion, which is significant for classifying objects by type and activity.

The third dimension has 168 functional attributes acquired during each time window, including real and imaginary parts, amplitude, phase, and other properties of the radar signal, enabling an insightful view of the moving, shape, and material properties of the target. This will allow for a proper training setup of machine learning models on object classification based on movement patterns or physical properties.

The dataset features hovering UAVs (e.g., Mavic series and Phantom 3 Professional) and a bionic bird, which are more difficult to detect due to small sizes of the radar cross section (RCS) and their composition in materials like carbon fiber and plastic that reflect a slight radar energy [27]. Additionally, hovering drones are more difficult to detect because they generate weak Doppler shifts. Simulated signals with overlaid noise will also further examine how pathologically close to natural such mimic systems could become.

## 4. Results

### 4.1. Introduction to the Results

This section describes the outcome of an investigation into the detection and identification of drones using machine learning algorithms applied to 60 GHz millimeter-wave radar data. The collected radar signals were analyzed with a bionic bird and two drones, i.e., DJI Mavic and DJI Phantom 3 Pro, under conditions in which artificial noises had been introduced. Major models like LSTM, GRU, CNN, and Transformer were also trained and evaluated based on the completed noisy dataset to establish performance metrics such as accuracy and error rates for degraded environments.

Full imitated realistic operating conditions included the full addition of white noise, Pareto noise, impulsive noise, and multipath interference. The boxplots present the data classification distribution, assuring demonstration of the algorithm’s capability in being able to classify between classes. The performance of the models was analyzed using ROC curves, and phase and amplitude data formed the prime input features for all of the models in this benchmark.

Furthermore, we extended the dataset to add two more statistical features—kurtosis and skewness—to improve detection and overcome the challenge of noise. Hence, we used Multimodal Transformer architecture, integrating them with amplitude and phase data. This enhancement resulted in better classification accuracies and robustness against noise, hence presenting a more efficient and reliable method for radar-based UAV detection. This result highlights the effectiveness of the proposed method, particularly in scenarios where robust performance is required despite the presence of noise.

### 4.2. Noisy Signal Visualization

This research discusses the effects of various noise patterns on mmWave radar signals, such as white noise, Pareto noise, impulsive noise, and multipath interference, which entails some parameters controlling their intensities and behaviors.

White noise is determined by the noise factor parameter, whose value is set to 10, indicating that it will multiply the standard deviations of the real and imaginary parts of the signal’s mean and add a Gaussian effect centered around the signal’s average.

For Pareto noise, the weight of the noise tail is defined by the alpha parameter with a value of 1.5, as well as by the noise factor, with a value of 10, which indicates the deformation of the intensity toward the relative correlativeness of the noise computed in relation to the standard deviation of the signal.

Impulsive noise was modeled with impulse ratio of 1, thereby indicating that about 100% of the samples will be affected, as well as scale factor of 10, which was used to define the amplitude of the impulsive noise.

Multipath interferense was defined using specifiers number of paths (10 reflections), maximum delay (10 samples), and maximum intensity (10) to simulate reflections that have different delays and intensities.

These noise models simulate real-world scenarios where interference occurs with the parameters chosen to reflect the average environment that radar systems might encounter.

As shown in Figure 2, for amplitude, the white noise showed a well-defined range of values, while its interquartile range appeared to be very narrow but was compensated with a significant number of high-magnitude outliers. These outliers signify that white noise contributed to the variability in the data, but most of the amplitude values were focused around the lower end. The presence of such extremes suggests that when signals become distorted in their dynamics, particularly in the amplitude domain, it would affect the later processing or modeling of data. Phase, as per the right boxplot, seemed to be less influenced when compared to the amplitude with white noise. The interquartile range was wide, with most values in a controlled range between approximately 1 and 3. Furthermore, there were no extreme outliers in the phase data, which indicates that the phase was relatively immune to white noise contaminations. Thus, white noise does not degrade the phase information but allows it to be preserved for certain applications.

For the Pareto noise, as shown in Figure 3, the amplitude (left) underwent drastic changes from the low levels of noise, as seen by a very broad range of values but also many outliers with very high magnitudes. Most of the data points tended to cluster around zero with an interquartile range of zero, which means that the noise had introduced a highly skewed distribution. This phenomenon aligns with known properties of Pareto noise, which can generate long tails with extreme deviations. The phase (right) seemed to not be disturbed by the Pareto noise so much. The interquartile range was well defined, with most values lying in an extremely limited range and having no visible outliers. This seems to indicate that at least the noise was affecting the phase a bit but, in comparison to the amplitude, its effects are mild. The limited distribution signifies that perhaps the phase data can yet be slightly interesting without going beyond what is needed for a model or an analysis.

The boxplots in Figure 4 show the effects of impulsive noise on the amplitude component (left) and phase component (right) of radar signals.

Amplitude-wise, impulsive noise causes high variability, with IQR clumped near low magnitudes, and the median is nearly to the bottom of the IQR. The most pronounced feature is denoting several extremely high outliers. These outliers show that impulsive noise severely deforms some amplitude data points, resulting in isolated spikes with high errors. Such behavior conveys that the impulsive noise contributes random but high amplitude deformations to the types of signals. On the contrary, the phase data were quite resistant to impulsive noise. The IQR was broad; most values lay between approximately 1 and 3 in error magnitude. The median phase error was located well within the IQR, thus indicating a more consistent distribution. More importantly, the nonappearance of extreme outliers in the phase data means that impulsive noise affects the phase more uniformly, so it does not lead to high deviation values. This is indicative of the robustness of the phase with regard to the sporadic high-intensity character of the impulsive noise when compared to the amplitude.

Under multipath interference, as shown in Figure 5, the IQR was thus widely spread for magnitude errors. The median was a bit below the center of the IQR, indicating a slight skew toward lower values. The whiskers indicate a large error magnitude. The fact that outliers were not visible indicates that most of the errors caused by multipath interference were certainly falling within this interval, leading to a monotonous but high-impact effect on the magnitude of the signal.

The phase errors demonstrated a completely opposite distribution. The IQR was also significantly below the magnitude error, with the midvalue situated right inside the IQR. It indicates a more centralized and fierce distribution of effect of the multipath interference on the phase component. Comparatively, phase error ranges beyond that defined by the IQR are indicated by the whiskers of the box, although the comparatively small size of the box suggests that such extreme deviations are rare events. Overall, a lack of extreme outliers suggests that multipath noise is rather gentle in introducing errors in the phase when compared to magnitude.

### 4.3. Comparative Analysis of the Algorithms

The performance of four algorithms, LSTM, GRU, Conv1D, and Transformer, was compared in two scenarios: the clean and noise-contaminated signals with four types of noises: white noise, Pareto noise, impulsive noise, and multipath interference. The trade-offs between true positive and false positive rates at different thresholds are shown in these results using ROC curves. The four models were then discussed in terms of how they were used to classify three dissimilar classes in the dataset: Mavic drones, Phantom 3 Pro drones, and bionic birds. These four architectures were expected to hold all comparisons between different approaches in deep learning, from sequences to attention-based models.

Table 1 summarizes the architectures of four machine learning models: LSTM, GRU, Conv1D, and Transformer. Long short-term memory (LSTM)—a recurrent neural network (RNN) that enables the model to capture temporal relationships in sequence data. In this architecture, there are two LSTM layers of 64 and 32 units, respectively, which then goes down to the dense output layer where softmax activation is performed to classify the input to one of the three classes. LSTMs can retain long-term dependencies and can analyze continuous signals over time. This is crucial for detecting subtle differences between signals that are very similar, such as between radar signatures of drones and birds. The differences may not be easily detected in small segments of data. However, LSTM models are more computationally extensive and may take longer to train than other architectures.

The GRU model is another variant of RNN like LSTM but has a far simpler structure than LSTM. The GRU implemented in this study had two layers of 64 and 32 units and a softmax output layer. The design of GRU addresses the problem of vanishing gradients in older versions of RNNs, but with far fewer numbers of parameters compared to LSTM, making it very cost-efficient and easy to train. In addition to saving time, the simplicity of the GRU model proved to be advantageous, particularly when the experiment being conducted involves large datasets or has lesser computational resources. Furthermore, the GRU model captures accurate temporal patterns of amplitude and phase data from drone and bird detection efficiently without overfitting. Generally, the results are comparable to using LSTM for this, but the lighter architecture provides quick experimentation and tuning for the task.

The Conv1D module takes a one-dimensional approach, utilizing convolutional layers to highlight the model’s particular strength in local pattern extraction. The model works simply by making use of two Conv1D layers—consisting of 64 and 32 filters, respectively, each with kernel size equal to 3—and it is followed by flattening and by a dense output layer being triggered by softmax. The Conv1D models lacked recurrence and use sliding convolutional filters rather than form patterns, which were identified by fixed-size windows in the input sequence. Here, we combined varying signal characteristics as they could indicate frequency changes in the radar signals. Conv1D forms excellent models for short-term variation detection, as expected from RNNs-only models, though they may not be able to learn longer-term dependencies. They may, however, be effective in detecting very specific local features that make one class slightly different from another.

Transformer, the most advanced architecture based on attention mechanism, has become worlds apart from conventional sequential architectures as it can simultaneously process minutes of parallel input. The architecture performs self-attention that distinguishes important features at different time steps. In the Transformer model that was implemented, there was sequential layer normalization followed by the multi-head attention layer, with 8 heads and a key dimension of 128. Further, the dropout layers were the regularizer, and the final output was obtained after passing an input through batch normalization and a 64-unit dense layer before sending it into a softmax layer. Multihop attention facility offers an enveloping context to the model for simultaneous attention on different aspects of radar signal, which makes the difference of it identifying various signatures of Mavic drones, Phantom 3 Pro drones, and bionic birds. The attention mechanism in this case also would signify that a given model could focus on the important portions of the input sequence, like the sudden spikes in amplitude or phase shifts, which could be critical for classification.

Table 2 summarizes the algorithmic complexity of the implemented models being measured: LSTM, GRU, Conv1D, and Transformer. The Time Complexity column indicates the computational expense of each model in sequence length (*n*) and hidden dimensions (*d*). Both LSTM and GRU were implemented according to the time complexity $O(n\cdot d^{2})$ with the property that they take as inputs iteratively during each time step, which made them ineffective when the length of the sequences increased. Conv1D, in comparison, had time complexity O(n·k·d), where *k* is the kernel size. It suffered significantly because the convolutions were conducted on the entire input sequence, rather than at individual time steps. The Transformer model, according to time complexity, assumes $O(n^{2}\cdot d)$ due to a self-attention mechanism that calculates pairwise interactions between all the tokens present in the input sequence.

For space complexity, all of the models can roughly store hidden states or attention weights with O(n·d) memory scaling and both the sequence length and internal dimensions of the model. The Sequential Dependency column shows a key difference between LSTM and GRU regarding high sequential dependency, meaning that each time step depends on the output of the preceding one, which precludes parallelization. By contrast, however, Conv1D and Transformer have no sequential dependency, which allows them to process input sequences in parallel, significantly improving efficiency in computation.

Finally, the column named Parallelization indicates that, though LSTM and GRU suffer from limited parallelization since they process data sequentially, both the Conv1D and Transformer models allow tremendous parallelism. Such characteristics make Conv1D and Transformer significantly more amenable to modern hardware such as GPUs, where parallel processing can significantly lower training time. Overall, this comparison illustrates that, although recurrent models may capture temporal dependencies more effectively, nonsequential models such as Conv1D and Transformer can provide better scalability and efficiency, particularly for large datasets, like the drone and bird detection dataset used in this research. This is a hands-on evaluation of models against predetermined baselines—it measures performance only with noise. The curves show that there are sizable divergences between these algorithms in terms of their behavior under different noise types, thus marking out each algorithm’s level of sensitivity to noise in signals.

An analysis of the boxplots (as shown in Figure 6) for classification probability under a white noise condition showed that the algorithms, namely LSTM, GRU, Conv1D, and Transformer, significantly differ in performance.

When looking for the ability of any of the algorithms to classify a given object (Bird, Mavic drone, or P3P drone, as shown in Figure 7), it shows low variability for those algorithms: LSTM, GRU, and Conv1D.

It also shows that their median probability values cluster near 0.4 or lower. These figures highlight the ineffectiveness of the algorithms in classification tasks as these values are not near those levels expected for highly confident predictions. Finally, there were also very close clusters with low variance, which shows that they have equally produced output that is not satisfactory in classification while under the white noise conditions. Transformer, however, displayed completely different behavior, characterized by wider interquartile ranges and higher probability medians for all three classes. This indicates that it is a significantly more robust and capable engine in dealing with the noise dataset than all the other algorithms; hence, the wider and more accurate range of output probabilities predicted.

The ROC curves of LSTM, GRU, and Conv1D appeared almost diagonal and had AUC values of 0.49–0.50 compositively across the three classes. This shows that the algorithms are not capable of distinguishing the classes because their depicting performance is like random guessing under white noise conditions. Transformer, however, had exceptionally high AUC values: 0.97 for Bird, 0.92 for Mavic, and 0.98 for P3P. This resulted in poorly inclined ROC curves for Transformer since their placement was not in the direction toward the top left corner. However, they remained quite steep toward said corner, which is indicative of its superior classification ability even with noise.

The boxplot in Figure 8 indicates a considerable difference in the probabilities assigned to class labels by different models and classes when the noise being considered is Pareto noise. Both LSTM and GRU seem to have higher median probabilities assigned to the ’bird’ class, indicating a better performance in this category.

Moreover, all of the classes of Transformer models performed fairly well because they showed a reduced number of outliers in the performance scores for all of the outputs.

These findings are further supported by ROC curves, where the Transformer model outperformed the rest of the models in terms of AUC scores by indicating better discriminative power across classes. Such Pareto noise definitely affects the performance of all models as it systematically decreased the overall accuracy and affected the positive side by increasing the number of false positives. This highlights the challenge these algorithms face in generalizing to noisy data. A potential area for future work could involve advancing modeling efforts to design more robust models, as stated in Figure 9.

The boxplot in Figure 10 shows the distribution of the classification probabilities across models and classes under impulsive noise, where the Transformer model consistently had a higher median probability, suggesting a higher confidence in predictions. The ROC curves in Figure 11 support these results.

The fact that the boxplot also shows wider ranges of probabilities for Transformer implies that, in some cases, it also has a higher level of uncertainty. In the results, typically, the Transformer model is better in terms of AUC values, with the major focus on Bird and Mavic classes, which mean there is better discrimination through Transformer among those classes with impulsive noise compared to the rest. The scenarios are similar for LSTM and GRU, although slightly lower in AUC than Transformer. The Conv1D model, however, performed poorly in most cases and particularly for the ’bird’ class, indicating that it is a convolutional architecture, which is not suitable for the temporal dependencies within the data.

Classification performance suffers from multipath interference; in particular, this was evident for LSTM, GRU, and Conv1D. The higher variability of classification probabilities, as seen in the boxplot, indicates that these models, while not infallible, display less sensitivity to the noise created from multipath. In comparison, Transformer shows more stability in performance as it is robust under these extreme conditions. All these claims are borne out by the ROC curves, where Transformer’s AUC scores were almost consistently superior to all the other models and can be considered the best under noisy conditions.

The boxplot in Figure 12 shows the data distribution of the classification algorithms considering multipath interference.

Figure 13 graphically shows the performance differences of the four machine learning algorithms when working on the classification of objects with multipath interference.

The Transformer model consistently outperformed LSTM, GRU, and Conv1D in terms of probabilities of classification and having a greater area under the ROC curve (AUC) value. This superlative performance is attributed to the extraordinary attention mechanism of Transformer in weighing the different parts of an input sequence and catching long dependencies. Both LSTM and GRU are wonderful in performing other tasks, however, and they have a considerable amount of confusion for multi-patterns created through multipath interference and, therefore, yield lower classification accuracies. Albeit, they were poorly designed for the temporal dependencies prevailing in radar data while the CNN model is optimal for spatial data.

### 4.4. Proposed Method for Enhanced Drone Detection Efficiency

Voluminous and complicated data from several millimeter-wave sensors was put through a methodology that functions efficiently to classify the drones at computation and efficiency, as well as accuracy. The noisy and complex radar signal dataset was processed into four distinct features: the amplitude and phase as real physical features of the radar signal that are more direct inferences from the signal’s inherent characteristics; and skewness and kurtosis because they are the derived statistical features that can capture higher-order moments of the distribution of signals.

The methods for extracting features from radar data are presented in Table 3.

Amplitude is obtained from the magnitude of the complex signal, and phase is computed from the angular component. The skewness and kurtosis computed along the temporal axis capture the asymmetry and peakedness of the signal distribution. All temporal and signal channel measurements were incorporated into the derivation of these features.

Then, the algorithm processed the multimodal data with Transformer after feature extraction. Each one of these feature maps was saved in independent arrays in order to keep amplitude, phase, skewness, and kurtosis as distinct representations. Each input had its corresponding sample label mapped to numeric values indicative of the class it fell under (Mavic, Phantom 3 Pro, or bionic bird).

The extracted features were then subject to the Z-score normalization, which normalized all the data to a zero mean and unit variance across all of the samples. Then, all the feature maps were fused along the last dimension to create a coherent input tensor. This final fused tensor was a vector consisting of the whole multidimensional input, which captures complementary aspects of the radar signal as input to the Transformer network for classification.

These features were the input features for the Multimodal Transformer model [50], which was developed for drones and artificial bird classification. Combining the physical and statistical characteristics of the input signals leads to increased classification accuracy and robustness.

The Multimodal Transformer function constructs a deep learning model for classifying time-series data that is produced by radar sensors in order to categorize their data as that of Mavic drones, Phantom 3 Pro drones, and bionic birds. There have been several modifications made to the Transformer architecture to increase its capability in handling and interpreting multimodal radar data such as amplitude, phase, kurtosis, and skewness. The goal was thus to analyze these features and detect and classify small aerial targets based on micro-Doppler signatures, which is difficult because of the similarity characteristics between the two types of targets.

First, this function depicted in Figure 14 defines an input layer with a shape compatible with the dimensions of the dataset. Input consists of radar-derived features capturing essential aspects of the motion of the target, such as as the amplitude and phase variations over time. These features help distinguish various targets because drones and birds have distinct motion patterns. The input data have been normalized using a LayerNormalization layer, which means that, essentially, the data will have a common distribution and keep fat tails from dominating learning. This stabilizes and hastens convergence by reducing internal covariate shift.

The core of multi-head attention lies in its ability to simultaneously focus on different parts of the input sequence. It specifies eight attention heads and a key dimension that determines the size of the query, key, and value vectors used in the attention calculations. The dropout rate of 0.1 adds a regularization effect to prevent overfitting, where some neurons are randomly deactivated during training. Such an attention mechanism helps this model capture more complex temporal dependence relations within those radar signals, which is crucial for distinguishing between targets, such as different drone models and bionic birds.

Subsequently, following the attention layer, and subjecting to a dropout rate of 0.2, the model further continued to prevent overfitting. The out-of-attention mechanism proceeded with a normalized output that meets a GlobalAveragePooling1D layer that summarizes information throughout the entire sequence to a fixed-size vector. This pooling operation guarantees that the global capture of the model in the data occurs while output keeps the dimension down so that easier processing can be performed by the next dense layers.

Next, the model entered a new phase consisting of two dense layers that constituted a feed-forward network through which the first, in total, comprised 256 neurons. The layer incorporates L2 regularization to overhaul the weights and prevent them from being excessively large. Here, the activation function in use is a LeakyReLU as it addresses the problem of dead neurons, which arises with standard ReLU processing, by letting negative values sag slightly to allow a small gradient to move through them. This means it suits radar data well, where models should learn both positive and negative variations as input. To impose further regularization, a dropout layer with a dropout rate of 0.3 was applied subsequent to the first dense layer. The second dense layer was constructed in an identical manner, containing 128 units and LeakyReLU activation, refining the feature representation further.

Thus, the last layer is a dense layer with softmax activation as its classification output probabilities for the respective target class: Mavic drone, Phantom 3 Pro drone, or bionic bird. Thus, the sum of the output values was found to be equal to 1 and they were treated as a multi-class classification problem. The model is compiled with Adam as the optimizer with a learning rate of 0.0001 to allow the model weight rearrangement during training, which has a sparse categorical cross-entropy as the loss function that is suitable for multi-class classification tasks with integer-encoded labels.

The aforementioned modifications were made in the Multimodal Transformer model with regard to optimizing the performance concerning detection of Mavic drones, Phantom 3 Pro drones, and bionic birds in RADAR data. These modifications are described in Table 4. Learning rate schedulers, early stopping, and class weights are some of these measures that come into play concerning certain layers and blocks of the model to ensure accurate and balanced predictions in the device’s different multidimensional classes.

First would be the Learning Rate Scheduler, which changes the learning rate dynamically during training, starting from a higher rate, thus allowing quick exploration of the parameter space, and then lowering toward the end for better convergence. Affected by this is the multiple head attention layers that work their magic in guaranteeing that the model receives relevant patterns of the radar signatures without instability. Furthermore, class weights are used to mitigate the class imbalance in the dataset, which is essentially critical for avoiding a biasing of the more represented classes and ensuring that the model trains freely for classifying Mavics, Phantom 3 Pro drones, and bionic birds.

The early stopping mechanism keeps track of the validation loss and interrupts the training after a specific point of not improving for five epochs. This is to avoid overfiting, thus maintaining the generalization capabilities of the models. Thus, dropout layers also benefit from this mechanism by regularizing without needing excessive training. The feed-forward network (FFN) was thus affected by class weights, which makes it strong in learning representations balanced across the classes.

The output layer using softmax for classification was influenced by class weights and early stopping. Hence, it ensures the probability distributions produced by the model are balanced and not biased toward any particular class, thus giving an overall improved performance in object detection across classes in the dataset.

These optimizations ensure Transformer effectively handles the complexities of the radar dataset, improving the detection accuracy of drones and birds, even under challenging conditions, such as noisy signals or similar movement patterns between different objects.

The model was trained for 15 epochs with a batch size of 64, and it used 10% of the data for validation. The performance metrics tracked the training and validation accuracy and loss.

The distribution of radar feature data, polluted with white noise (Figure 15), shows a much narrower and more compact distribution for the Mavic and Bird classes, with less dispersion, and it shows a wider data distribution for the drone P3P class.

The original structure of the signal was maintained, which most likely facilitates the discrimination between the classes; consequently, a better classification is performed by the algorithm, as can be seen in Figure 16.

For Pareto noise, which was previously our critical case, even with Transformer, there were a couple of more visible outliers, at least in the amplitude and kurtosis measures, but the main bulk of distribution, as represented in the boxplot (Figure 17) through the interquartile ranges and medians, remained very similar across the classes.

Pareto noise had slightly positively skewed distribution characteristics when compared to impulsive noise, but such a difference did not tend to produce anything remarkably dissimilar in the overall similarity between classes, as shown in Figure 18.

Based on the graphs analyzed in Figure 19, the boxplots for impulsive noise revealed similar distributions among the classes Bird, Mavic, and P3P in all the evaluated metrics (amplitude, phase, skewness, and kurtosis). There were no significant differences in the central values, dispersion, or presence of outliers indicating any substantial changes caused by impulsive noise.

This patchiness is not strong enough with respect to the effect of impulsive noise on the data—at least in reference to those metrics—to distinguish patterns that overwhelmingly classify the data into groups (Figure 20).

The result in Figure 21 indicates perfect separability despite the uniform signal distortion caused by multipath effects.

The Multimodal Transformer model demonstrated excellent performance in distinguishing between the Bird, Mavic, and P3P classes under multipath interference (Figure 22), achieving an AUC of 1 for all classes. A similarly high performance was observed under white noise and impulsive noise conditions, confirming the robustness of the model across different noise types. However, the classification accuracy was slightly lower under Pareto noise, highlighting that heavy-tailed noise distributions pose a greater challenge to the model. Overall, Transformer effectively handles various noise interferences, maintaining reliable classification even under complex signal distortions.

The Multimodal Transformer algorithm, indicated by the above complexity forms, followed the same time complexity logic as Transformer, with slight differences. The attention operation, at this stage, was essentially going to incur a time complexity or storage requirements that are quadratic with respect to the sequence length in the self-attention mechanism because it, in actuality, takes part in all dimensions. This implies that the time complexity is $0\;(n^{2}cd)$, *n* in this case, when it really refers to the length *c* of the sequence and *d* according to the dimension of the word representations extracted features. This means that the associated space complexity is also about 0 (ncd) since the intermediate representations need to be eliminated as the model will also have a need for storing attention weights.

However, even if the normalization, pooling, and dense layers with regularization have no impact on the order of magnitude as far as complexity is concerned, they do mold the specific implementation behavior. As far as parallelization is concerned, the implementation would be considered highly parallelizable since the Multimodal Transformer processes nearly all incoming information input in parallel (due to the main usage of the MultiHeadAttention mechanism). Therefore, it can execute this on modern hardware like GPUs.

## 5. Discussion

The excellent performance exhibited by the Multimodal Transformer model in distinguishing drone and bird classes showed that there are several factors that are crucial in its robustness against noise interference. One important attribute is the use of attention mechanisms as these allow the model to only refer the input data to the relevant parts, thus reducing the effect of impact from noise. Features such as amplitude, phase, skewness of amplitude, as well as the kurtosis of amplitude, are critical in retaining significant signal characteristics. These features still remain informative despite the introduction of white noise, Pareto noise, impulsive noise, or multipath interference, and they enable Transformer to achieve near perfect classification.

The architecture of Transformer is structurally built to capture complex non-linear relationships between features that may not be captured by other models. Unlike traditional methods that may be considered biased toward simple or localized patterns, the mechanisms of Transformer tend to weigh the input features on the entire set. This flexibility allows the system to detect those subtle features embedded in the data, which is important for accurate classification, even under noisy conditions.

In addition, there is a self-embedded feature of the model of extraction and ranking of critical information from variously input characteristics so that the irrelevant noise does not overpower the important parts of the signal. This then depicts the noise resilience of Transformer, showing its propensity for intricate feature extraction and analytical interpretation—an invaluable asset for classifying radar signals in challenging real-world environments.

The robustness demonstrated by Transformer also highlights its potential for practical applications in domains such as public safety and military defense. In public safety, the model could be employed to detect unauthorized drones in sensitive areas such as airports, government buildings, or public events. Similarly, in military scenarios, Transformer could be used to identify and classify drones in complex and hostile environments. However, deploying Transformer in real-world operations presents challenges, especially concerning computational requirements for real-time analysis and the need for continuous model updates to adapt to evolving threats and conditions.

## 6. Conclusions

This study demonstrates the Multimodal Transformer model as performing exceptionally in classifying drone and bird signals with different types of noise such as white noise, Pareto noise, impulsive noise, and multipath interference without any real degradation in the accuracy. This superior performance of the model is mainly because the attention mechanisms are effectively used to focus on the most relevant features in data that are significantly affected by noise. The features that were considered were amplitude, phase, skewness of amplitude, and kurtosis of amplitude, and they were used to thoroughly cover the characteristics of the signals that Transformer takes to class description.

Noise is, of course, a big problem, yet the very high AUCs of all the noise types demonstrate how robust the Multimodal Transformer model is in handling complex and noisy input data. However, the additional capability of the model to include both statistical ones, namely skewness and kurtosis, and physical ones, that is, amplitude and phase, is expected to contribute to considerably capturing subtle but meaningful patterns and improving generalization beyond immediate requirements. This trait becomes more important in real-world practical use cases where radar and sensor data are invariably noisy due to various sources, as seen in aerial object detection.

From these findings, it can be inferred that the architecture of Transformer can most likely fulfill the need for noisy radar signal classification. Further research could test more feature combinations in the model or the model could be optimized further to face other operational issues. The results laid a solid foundation for improving the reliability and efficiency of automatic drone detection systems in environments of different and highly noisy conditions.

## Figures and Tables

**Figure 1 sensors-25-00721-f001:**
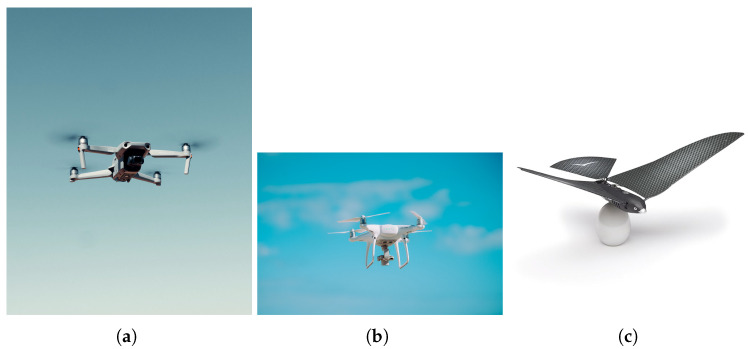
Images of the test subjects used in the experiments: (**a**) the DJI Mavic drone [47], (**b**) the DJI Phantom 3 Pro drone [48], and (**c**) the Bionic Bird [49]. These devices illustrate what were used to evaluate the classification performance of the Multimodal Transformer model.

**Figure 2 sensors-25-00721-f002:**
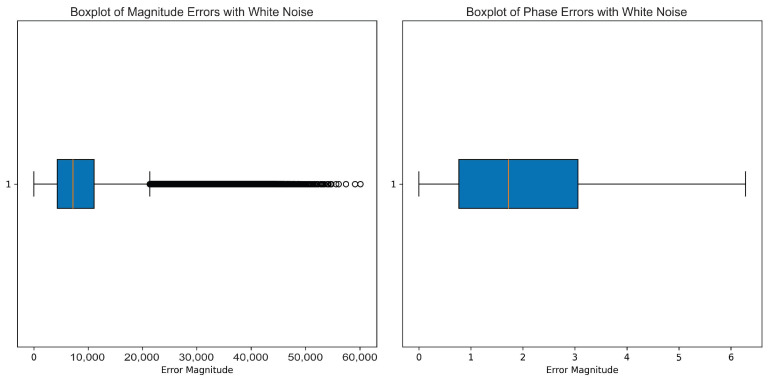
This boxplot shows the impact of white noise on the amplitude (**left**) and phase (**right**) of the radar signal. White noise follows a random distribution, primarily affecting the outliers in both amplitude and phase. The amplitude exhibits a broader spread, with more pronounced outliers in both directions. The phase is also impacted, though to a lesser extent, showing a slight median shift and a moderate interquartile range expansion.

**Figure 3 sensors-25-00721-f003:**
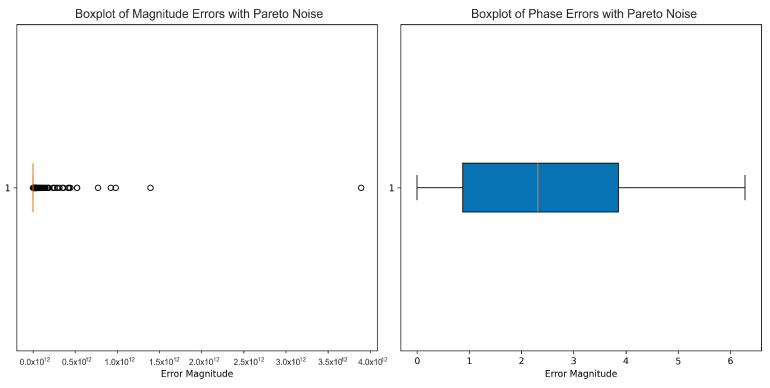
This boxplot illustrates the impact of Pareto noise on the amplitude (**left**) and phase (**right**) of the radar signal. Pareto noise, also known as heavy-tail noise, introduces extreme values more frequently than white noise, resulting in greater data dispersion. The amplitude plot shows a considerable number of high-value outliers, suggesting that the noise causes more frequent positive fluctuations. The phase remains relatively stable, with occasional extreme values.

**Figure 4 sensors-25-00721-f004:**
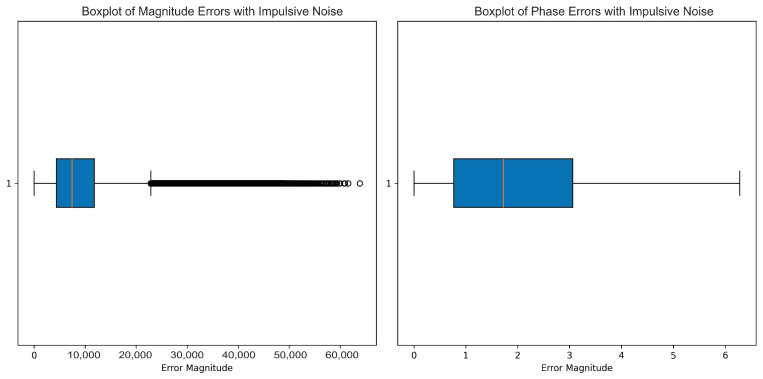
This boxplot illustrates the effect of impulsive noise on the amplitude (**left**) and phase (**right**) of the millimeter-wave radar signal. Impulsive noise generates abrupt, random spikes, causing significant data dispersion. The amplitude plot shows a noticeable increase in outliers at both extremes, with a wider interquartile range. While the median amplitude remains relatively stable, the data’s extremes were widely scattered. The phase plot shows a similar pattern, with more visible outliers and a slight median shift.

**Figure 5 sensors-25-00721-f005:**
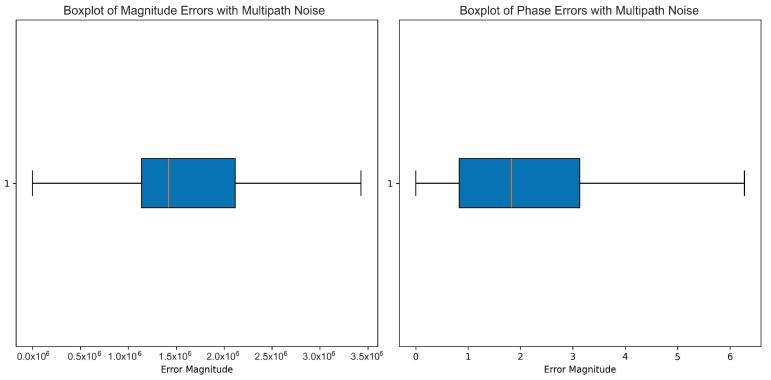
This boxplot shows the effects of multipath interference noise on the amplitude (**left**) and phase (**right**) of the radar signal. Multipath interference occurs when the signal reflects off multiple surfaces before reaching the receiver, causing distortions. The amplitude plot reveals increased variability and a larger number of outliers, indicating inconsistencies in the measured values. The phase is less affected but still shows a slight increase in dispersion compared to the original signal.

**Figure 6 sensors-25-00721-f006:**
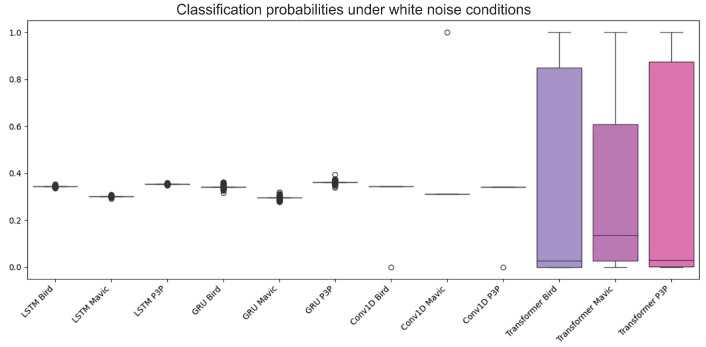
This boxplot illustrates the classification probability outputs of four algorithms—LSTM, GRU, Conv1D, and Transformer—under white noise conditions. The boxplot reveals that LSTM, GRU, and Conv1D exhibited tightly clustered probability distributions with low variance, and their median probabilities remained around or below 0.4 across all classes (Bird, Mavic drone, and P3P drone). This low variability and clustered median values suggests poor classification performance, with predictions lacking high confidence and distinguishing power. In contrast, the Transformer algorithm demonstrated a markedly different behavior, with wider interquartile ranges and higher median probabilities for all classes. The wider spread indicates that Transformer is more resilient to white noise, producing more varied and accurate probability outputs, thus highlighting its superior robustness in handling noisy data compared to the other models.

**Figure 7 sensors-25-00721-f007:**
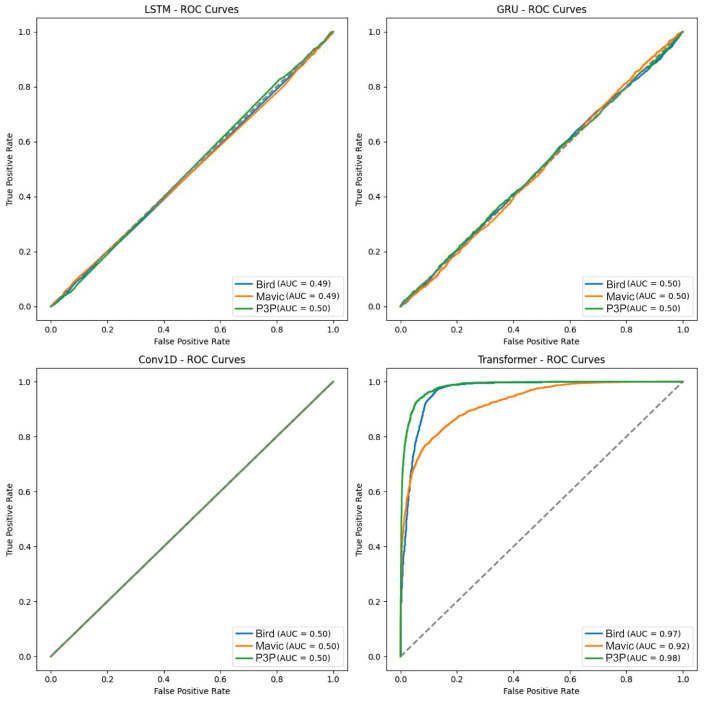
This boxplot presents the classification probabilities of four machine learning models—LSTM, GRU, Conv1D, and Transformer—under white noise conditions for the three target classes: Bird, Mavic drone, and P3P drone. The LSTM, GRU, and Conv1D models display tightly grouped probability distributions with narrow interquartile ranges and median values clustered near or below 0.4 across all classes. This indicates that these models struggle to produce confident predictions in noisy environments as their output probabilities remain low and exhibit limited variability, suggesting a uniform inability to distinguish between the classes under these conditions. In contrast, the Transformer model showed a significantly wider interquartile range and higher median probability values for all classes. This broader distribution highlights Transformer’s superior robustness to white noise, enabling it to generate more confident and diverse predictions across the dataset, outperforming the other models in terms of classification reliability under challenging noise conditions.

**Figure 8 sensors-25-00721-f008:**
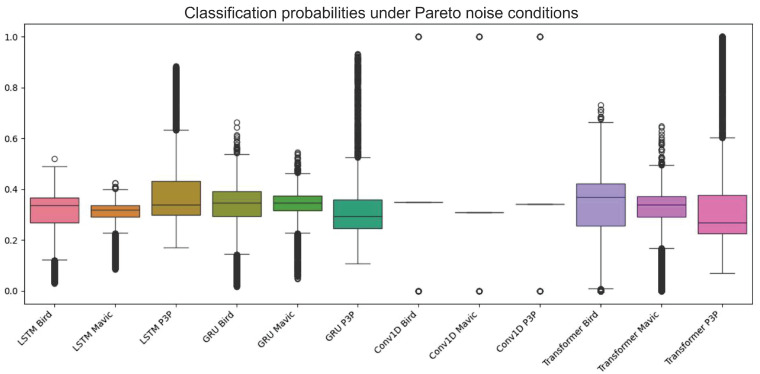
This boxplot illustrates the classification probabilities for different models—LSTM, GRU, Conv1D, and Transformer—under Pareto noise conditions across three target classes: Bird, Mavic drone, and P3P drone. The LSTM and GRU models exhibited higher median probabilities for the “bird” class, suggesting better performance in this specific category compared to other classes. However, the overall performance across all models was negatively impacted by Pareto noise, which introduces frequent extreme values (outliers) and disrupts the models’ ability to confidently assign accurate probabilities, particularly those affecting classification consistency.

**Figure 9 sensors-25-00721-f009:**
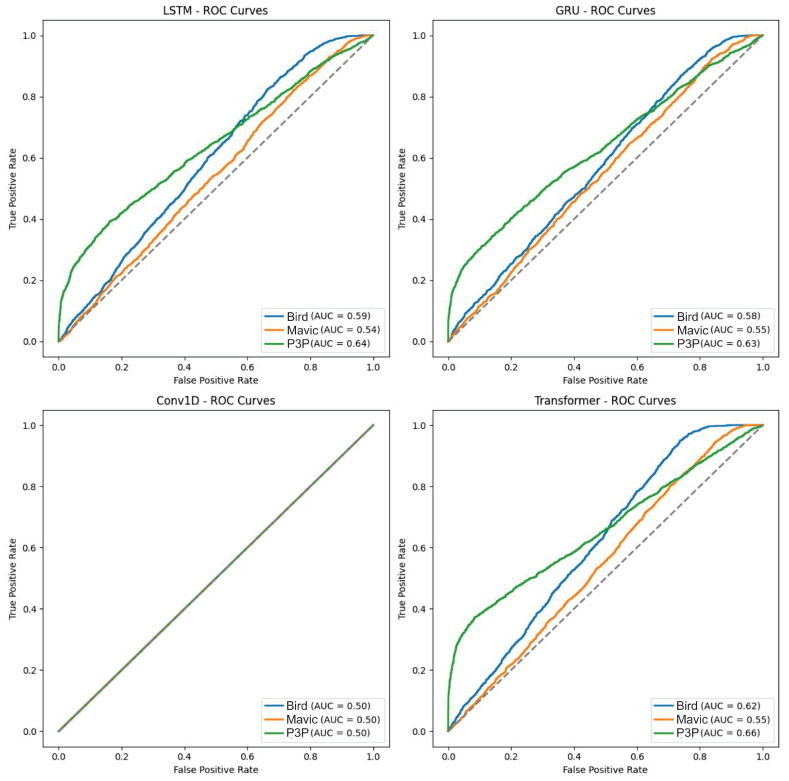
This figure presents the ROC curves and AUC scores for the classification performance of LSTM, GRU, Conv1D, and Transformer models under Pareto noise conditions. The Transformer model demonstrated superior performance, with higher AUC scores across all target classes, indicating better discriminative ability compared to the other models. Additionally, Transformer exhibited fewer outliers in classification scores, highlighting its robustness to Pareto noise. In contrast, the LSTM, GRU, and Conv1D models showed higher false positive rates, suggesting difficulty in generalizing to data with high variability caused by noise. These results emphasize the need for further model optimization to handle noise-induced challenges effectively.

**Figure 10 sensors-25-00721-f010:**
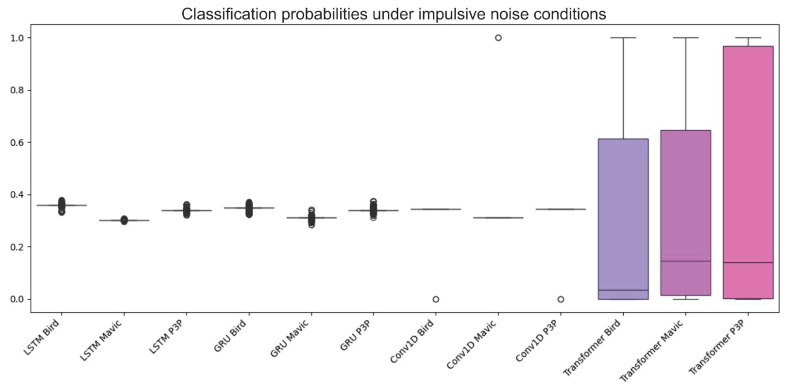
This boxplot displays the distribution of classification probabilities for the LSTM, GRU, Conv1D, and Transformer models under impulsive noise conditions across the Bird, Mavic drone, and P3P drone classes. The Transformer model consistently showed a higher median probability across all classes, indicating more confident predictions. However, its wider interquartile range suggests that it also exhibits greater uncertainty in some predictions. In contrast, the other models—LSTM, GRU, and Conv1D—showed lower median probabilities and tighter ranges, indicating less confidence and lower variability in their predictions under impulsive noise conditions.

**Figure 11 sensors-25-00721-f011:**
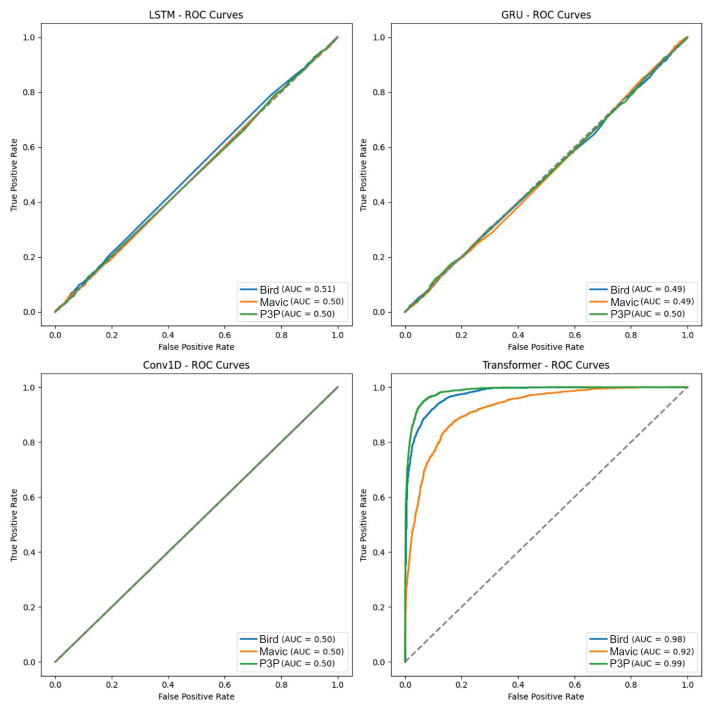
The ROC curves and AUC values demonstrate the classification performance of LSTM, GRU, Conv1D, and Transformer models under impulsive noise conditions. The Transformer model outperforms the other models, particularly for the Bird and Mavic drone classes, with higher AUC values, indicating better discrimination capabilities. The LSTM and GRU models show moderate performance but are slightly less effective than Transformer. The Conv1D model performs poorly across most classes, especially for the Bird class, reflecting its inability to effectively handle temporal dependencies in the presence of impulsive noise.

**Figure 12 sensors-25-00721-f012:**
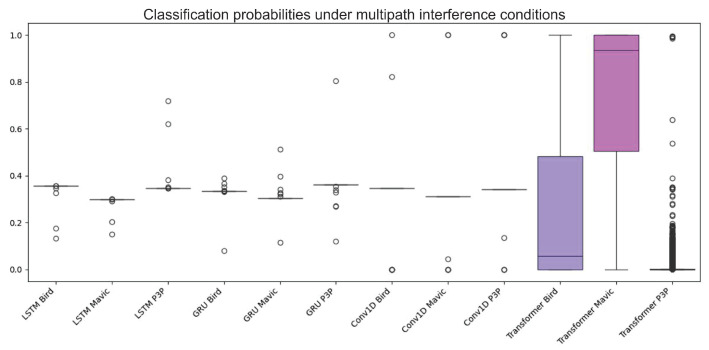
Boxplot illustrating the classification accuracy variability of the four machine learning algorithms under multipath interference. The Transformer model consistently demonstrated higher accuracy and lower variability, indicating superior stability and performance compared to the LSTM, GRU, and Conv1D models.

**Figure 13 sensors-25-00721-f013:**
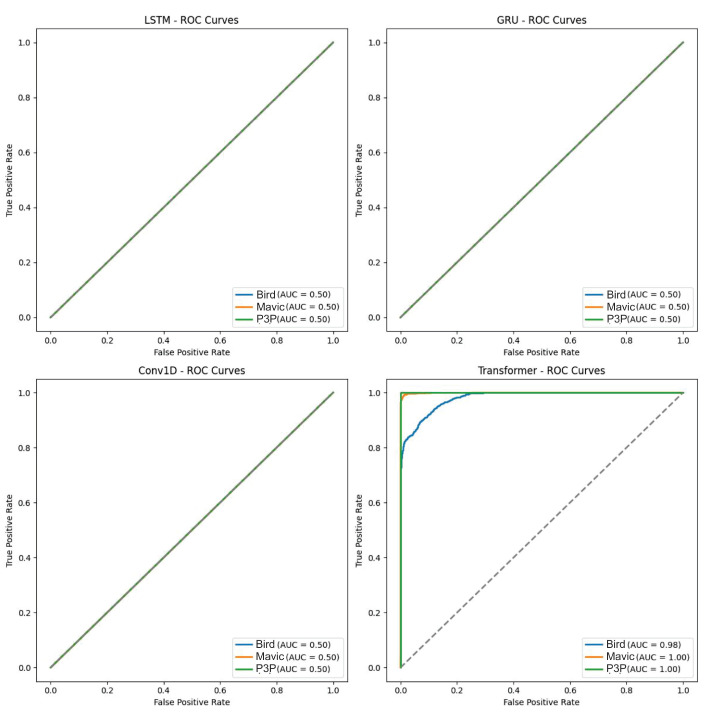
ROC curves showing the performance of the Transformer, LSTM, GRU, and Conv1D models in object classification with multipath interference. The Transformer model consistently outperformed the others, with higher AUC values, highlighting its robustness and superior attention mechanism for handling noise and temporal dependencies.

**Figure 14 sensors-25-00721-f014:**
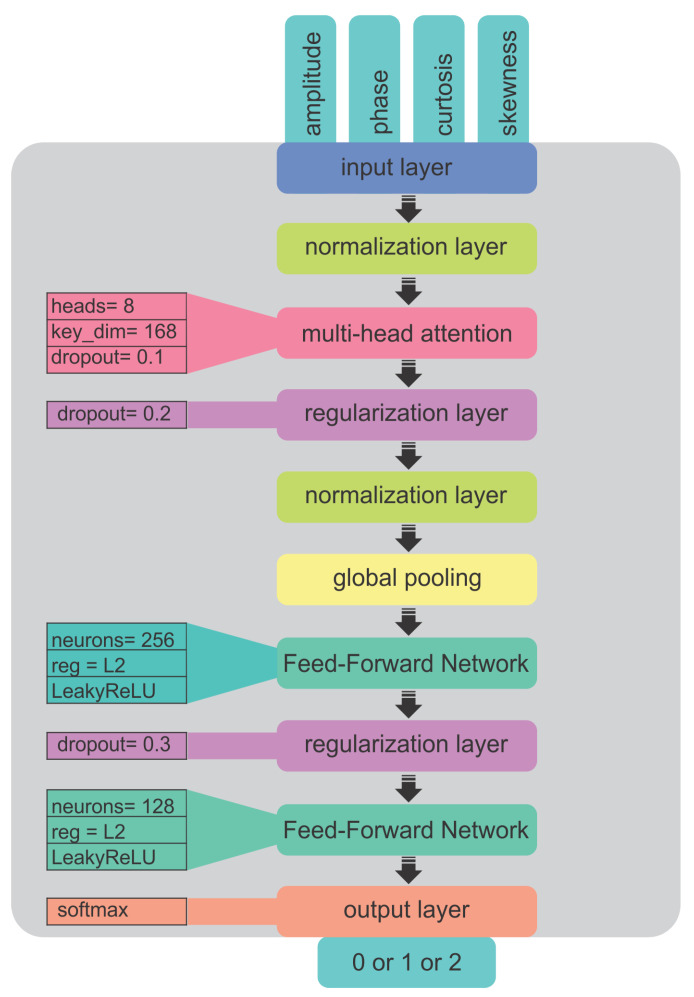
Schematic of the Multimodal Transformer Model (MMT) used for radar-based target classification. The model begins with an input layer processing radar features, followed by LayerNormalization for stable learning. Multi-head attention (8 heads) captures complex temporal dependencies in radar signals. Dropout layers (0.1 and 0.2) prevent overfitting. A GlobalAveragePooling1D layer reduces dimensionality, followed by two dense layers with L2 regularization and LeakyReLU activation. The final dense layer outputs classification probabilities using softmax, where the model’s output is 0 for the Mavic drone, 1 for the Phantom 3 Pro drone, or 2 for bionic bird. The model was optimized with Adam and sparse categorical cross-entropy loss for multi-class classification.

**Figure 15 sensors-25-00721-f015:**
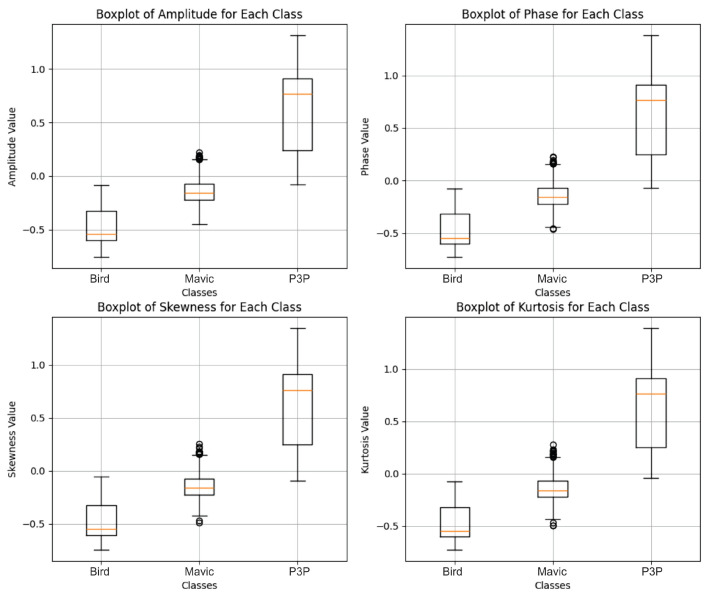
Boxplots showing the amplitude, phase, skewness, and kurtosis values for the Bird, Mavic, and P3P classes, as extracted from radar signals with added white noise. The Bird class showed lower and more stable values across all features. The Mavic class had moderate values with noticeable outliers. The P3P class consistently showed the highest medians and broader ranges, indicating stronger and more variable radar reflections. The differences in these features help to distinguish the classes in the Transformer model’s classification process.

**Figure 16 sensors-25-00721-f016:**
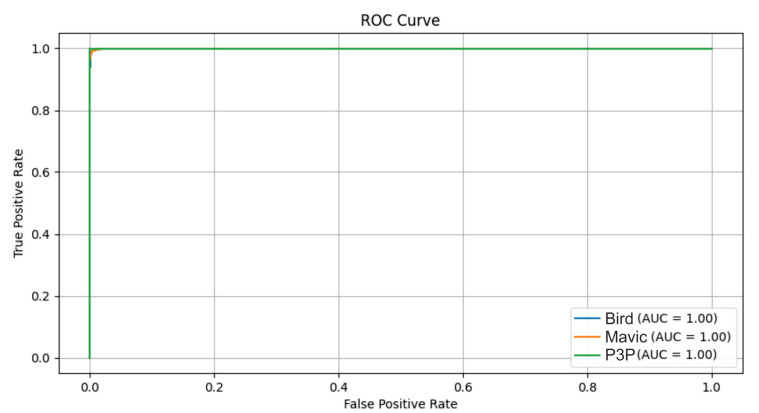
ROC. curves for the classification of the Bird, Mavic, and P3P classes. The model achieved a perfect AUC for all the classes in both noise cases. For white noise, the curves were slightly farther from the vertical axis compared to the Pareto noise, indicating a slightly better robustness to the latter noise type.

**Figure 17 sensors-25-00721-f017:**
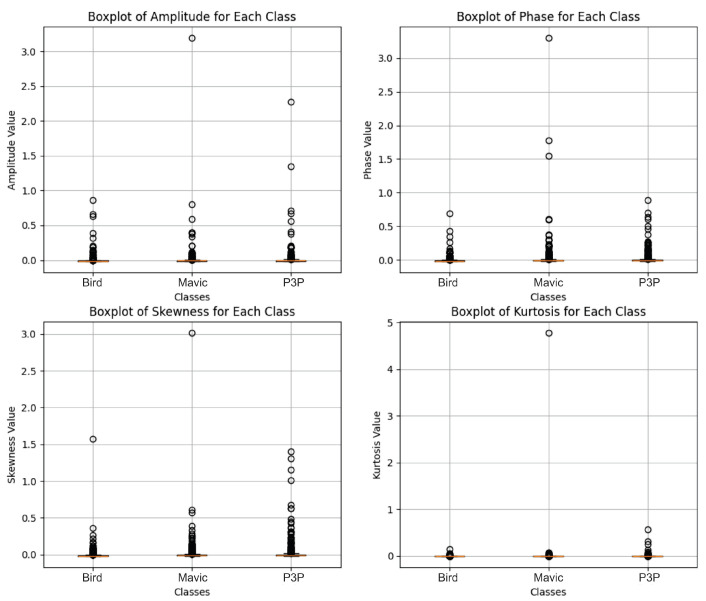
Boxplots showing the amplitude, phase, skewness, and kurtosis values for the Bird, Mavic, and P3P classes, as extracted from radar signals with added Pareto noise. The median values remained close to zero across the features, with fewer outliers and lower variability compared to white noise. The Bird class showed the most stable distribution, while the Mavic and P3P classes exhibited moderate spreads with fewer extreme values. This reduced variability under Pareto noise led to diminished class separability, resulting in lower classification performance compared to white noise.

**Figure 18 sensors-25-00721-f018:**
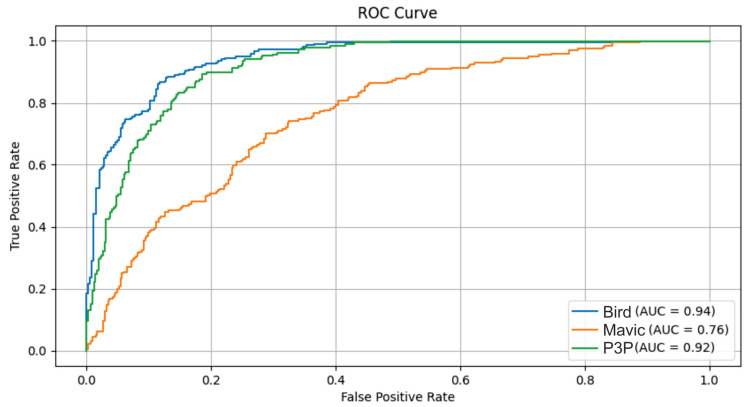
The ROC curves for the classification of Bird, Mavic drone, and P3P drone when considering Pareto noise with a Multimodal Transformer.

**Figure 19 sensors-25-00721-f019:**
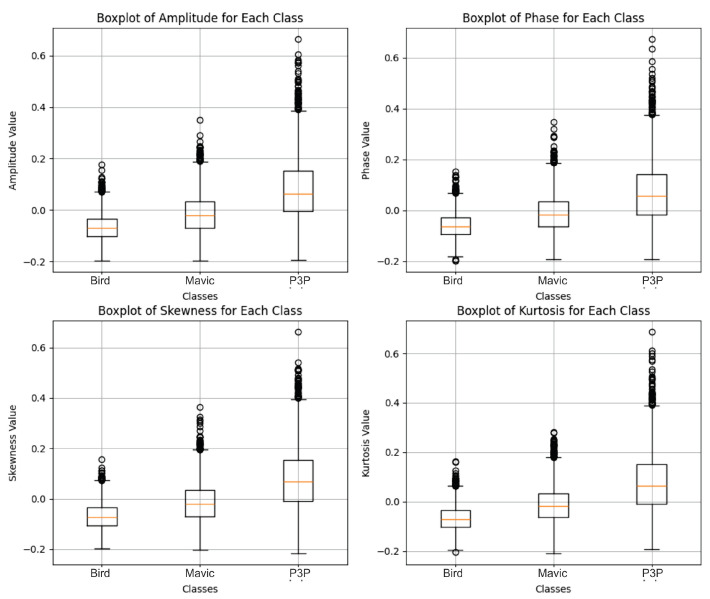
Boxplots showing the amplitude, phase, skewness, and kurtosis values for the Bird, Mavic, and P3P classes under impulsive noise. The Bird class exhibited the most stable and compact distribution across all features, while the P3P class showed the highest median values and widest variability, especially in amplitude and kurtosis. The Mavic class displayed intermediate behavior. Impulsive noise significantly increased outliers, particularly in the P3P class, indicating that larger or more complex targets produce more erratic radar reflections.

**Figure 20 sensors-25-00721-f020:**
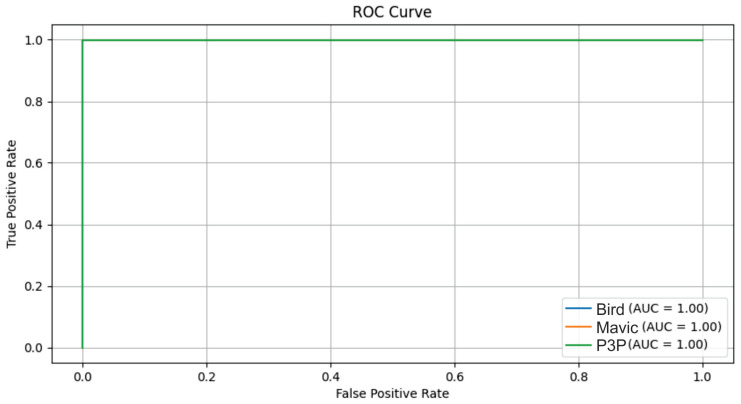
ROC curves for the classification of the Bird, Mavic drone, and P3P drone classes when considering impulsive noise with the Multimodal Transformer.

**Figure 21 sensors-25-00721-f021:**
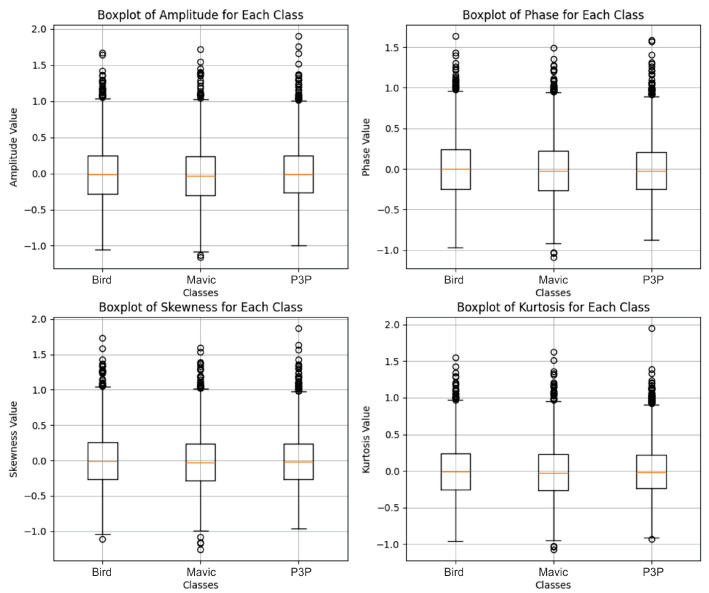
Boxplots showing the amplitude, phase, skewness, and kurtosis values for the Bird, Mavic, and P3P classes under multipath interference. Unlike impulsive noise, the distributions were more uniform across classes, with median values close to zero and consistent interquartile ranges. The P3P class showed a slightly wider spread in amplitude and skewness, suggesting higher susceptibility to multipath effects. Outliers were evenly distributed across classes, indicating random variations in signal reflections that reduce separability between classes.

**Figure 22 sensors-25-00721-f022:**
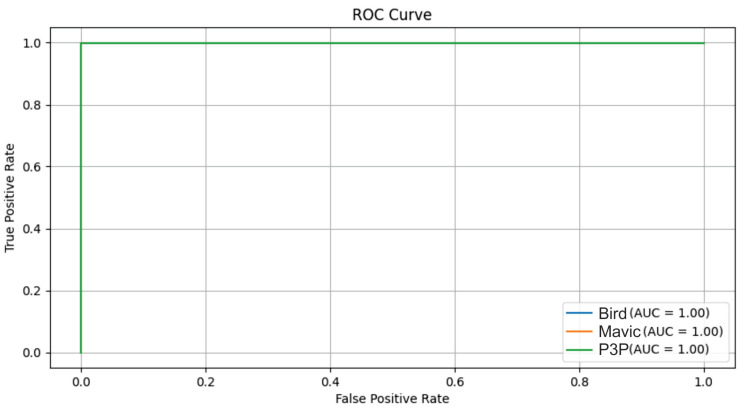
ROC curves for the classification of Bird, Mavic drone, and P3P drone classes when considering multipath interference with the Multimodal Transformer.

**Table 1 sensors-25-00721-t001:** Implementation details of LSTM, GRU, Conv1D, and Transformer models.

Model Type	Architecture Details	Input Features and Objective
LSTM	Two LSTM layers: - 64 units with return sequences - 32 units Output layer: Dense with softmax activation	Input: Amplitude and phase Objective: Capture temporal dependencies in sequential data
GRU	Two GRU layers: - 64 units with return sequences - 32 units Output layer: Dense with softmax activation	Input: Amplitude and phase Objective: Reduce computational cost compared to LSTM while retaining performance
Conv1D	Two Conv1D layers: - 64 filters with kernel size 3 - 32 filters with kernel size 3 Flatten layer Output layer: Dense with softmax activation	Input: Amplitude and phase Objective: Extract local patterns and features from sequential data
Transformer	Layer normalization, Multi-Head Attention (8 heads, key dimension 128), Batch Normalization, Dropout (0.2), Global Average Pooling, Dense layer (64 units), Output layer: Dense with softmax activation	Input: Amplitude and phase Objective: Leverage self-attention mechanisms for better feature extraction and pattern recognition in sequences

**Table 2 sensors-25-00721-t002:** Algorithmic complexity of the LSTM, GRU, Conv1D, and Transformer models.

Model	Time Complexity	Space Complexity
LSTM	$O(n\cdot d^{2})$ Sequential processing at each time step High computational cost for long sequences	O(n·d) Memory for storing hidden states
GRU	$O(n\cdot d^{2})$ Sequential processing at each time step High computational cost for long sequences	O(n·d) Memory for storing hidden states
Conv1D	O(n·k·d) Parallel processing across input sequence Lower computational cost	O(n·d) Memory for storing feature maps
Transformer	$O(n^{2}\cdot d)$ Quadratic scaling due to self-attention mechanism Pairwise interaction computation between all tokens	O(n·d) Memory for storing attention weights and hidden states

**Table 3 sensors-25-00721-t003:** Extracted features from the radar data and their computation methods.

Feature	Description	Computation Method
Amplitude	Represents the magnitude of the radar signal, indicating the strength of the reflected wave	|x| (absolute value of the complex number)
Phase	Captures the angular component of the radar signal, which provides information on the relative position of the target	∠x (angle of the complex number)
Skewness	Measures the asymmetry of the amplitude distribution, helping to detect anomalies in the signal	$\frac{E[{\left(X-\mu \right)}^{3}]}{\sigma^{3}}$ (third standardized moment)
Kurtosis	Quantifies the peakedness of the amplitude distribution, indicating how heavy the tails of the distribution are	$\frac{E[{\left(X-\mu \right)}^{4}]}{\sigma^{4}}-3$ (fourth standardized moment)

**Table 4 sensors-25-00721-t004:** Impact of the training adjustments on the Multimodal Transformer model.

Model Block	Adjustment Applied	Expected Benefit
Input Layer	Class Weights	Balanced predictions across all classes
Multi-Head Attention	Learning Rate Scheduler	Stable convergence and improved pattern detection
Regularization Layer	Early Stopping	Reduced overfitting and better generalization
Feed-Forward Network	Class Weights & Learning Rate Scheduler	Improved representation of underrepresented classes
Output Layer	Class Weights & Early Stopping	Balanced predictions and optimal stopping point

## Data Availability

The data supporting the findings of this study are available upon reasonable request from the corresponding author. Due to the nature of the research, some data may be restricted for confidentiality or ethical reasons.

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
