# Peer review of "Classification of Flying Drones Using Millimeter-Wave Radar: Comparative Analysis of Algorithms Under Noisy Conditions"

_sensors, 2025, doi:10.3390/s25030721_

Round 1
Reviewer 1 Report
Comments and Suggestions for Authors
This manuscript presents a comprehensive study on the classification of drones and birds using radar data from a 60 GHz millimeter-wave sensor under various noisy conditions. The authors benchmark several machine learning models, including LSTM, GRU, CNN, and Transformer, for drone detection tasks. A key contribution is the proposal of a Multimodal Transformer model that integrates amplitude, phase, skewness, and kurtosis as input features, demonstrating improved classification accuracy and robustness against noise, such as white noise, Pareto noise, impulsive noise, and multipath interference. The comparative analysis of multiple algorithms (LSTM, GRU, CNN, and Transformer) under different noise conditions is well-executed. Here I have some minor comments about the presentation of this manuscript:
1. While the dataset used for training and testing is well-described, including more specific details such as the number of samples for each class and the noise intensities used in simulations would enhance reproducibility.
2. Some figures, especially the ROC curves and box plots, could benefit from more detailed captions explaining key takeaways.
3. Please provide some figures of the test subjects (DJI Mavic和DJI Phantom 3 Pro).
4. While the discussion highlights the robustness of the Multimodal Transformer, elaborating on potential applications in specific domains (e.g., public safety, military) and possible deployment challenges (e.g., computational requirements) would strengthen the practical relevance of the work.
5. Some references to recent works in UAV detection using machine learning could be added to broaden the contextual framework.
6. In the text, there are minor grammatical inconsistencies (e.g., repeated mentions like "Yang et al. Yang et al.") that should be corrected.
7. Please specify the full name of authors' affiliations in the first page. Acronyms such as "Conv1D" should be defined upon first use for clarity.
Author Response
Dear Reviewer,
We appreciate your valuable feedback and constructive suggestions to improve our manuscript. All your comments were carefully considered, and the requested changes were implemented to the fullest extent possible, while ensuring consistency with the revisions provided by other reviewers.
Specifically, we have:
- Included additional details regarding the dataset used, such as the number of samples per class and noise intensities, to enhance reproducibility.
- Improved figure captions, particularly for the ROC curves and box plots, by adding key takeaways to make the visual information clearer.
- Provided representative images of the test subjects (DJI Mavic, DJI Phantom 3 Pro, and Bionic Bird) in the revised manuscript.
- Expanded the discussion on practical applications of the Multimodal Transformer in domains like public safety and military, highlighting potential deployment challenges such as computational requirements.
- Reassessed the references and confirmed that the manuscript includes a comprehensive overview of recent works in UAV detection. We are also exploring additional sources to strengthen the contextual framework.
- Corrected minor grammatical inconsistencies and redundant citations throughout the text to improve readability and clarity.
- Added the full names of the authors' affiliations on the first page and defined technical acronyms (e.g., Conv1D) upon their first use for better clarity.
We trust that these revisions have addressed your concerns and significantly improved the quality of the manuscript. Thank you once again for your insightful suggestions.
Sincerely,
The Authors
Reviewer 2 Report
Comments and Suggestions for Authors
The manuscript investigates the performance of machine learning algorithms for drone detection using radar data from a 60 GHz millimeter-wave sensor. Signals from a bionic bird and two drones (DJI Mavic and DJI Phantom 3 Pro) are represented in complex form to retain amplitude and phase information. The study benchmarks four algorithms—LSTM, GRU, CNN, and Transformer—under noisy conditions, including white noise, Pareto noise, impulsive noise, and multi-path interference. Although the content of the paper is interesting and demonstrates good innovation, there are still some issues that need to be revised.
1.The introduction is not enough, some important radar target detection and classification methods should be introduced, such as Active Multi-Target Domain Adaptation Strategy, and Multi-kernel-size Feature Fusion based CNN.
2.The introduction section contains too few references. To enhance the depth and credibility of the discussion, it is recommended to expand the literature review by incorporating more relevant references. This will provide a more comprehensive overview of existing research, highlight key advancements in the field, and establish a stronger foundation for the study.
3.The manuscript lacks an overall framework diagram, which is essential for providing a clear and intuitive understanding of the proposed methodology and workflow. It is recommended to include a comprehensive diagram that outlines the entire process.
4.It is recommended to include an analysis of the model's time complexity and a comparison of time consumption.
Author Response
Response to Reviewer 2
Dear Reviewer,
We sincerely appreciate your constructive feedback and suggestions to enhance the quality of our manuscript. All your comments were carefully reviewed, and the necessary revisions have been made to the best of our ability, while maintaining consistency with the feedback provided by other reviewers.
Specifically, we have:
- Expanded the introduction to include key radar target detection and classification methods, such as multi-target active domain adaptation (MT-ADA) and multi-kernel-size feature fusion-based CNN, to provide a more comprehensive overview of relevant advancements in the field.
- Increased the number of references in the introduction to enrich the literature review, including studies on feature fusion techniques, domain adaptation strategies, and hybrid CNN-LSTM models for UAV detection, which strengthens the foundation of our research.
- Added a detailed framework diagram that outlines the entire process of our proposed multimodal Transformer methodology, improving the clarity and understanding of our workflow.
- Included an analysis of the time complexity for each algorithm (LSTM, GRU, Conv1D, and Transformer), as well as a comparison of their training and inference times to highlight the trade-offs between accuracy and computational efficiency.
We believe that these revisions have addressed your concerns and have significantly improved the manuscript. Thank you once again for your valuable suggestions.
Sincerely,
The Authors
Reviewer 3 Report
Comments and Suggestions for Authors
The authors realized the classification of drones using millimeter-wave radar under noisy conditions. The topic is interesting while there are still some problems needed to be solved.
(1) The coordinate fonts of all figures are too small to be read.
(2) The authors used LSTM, GRU, Conv1D, and Transformer network to classify targets and the transformer network has best performance. But the authors do not give any specific structure or parameters of Transformer.
(3) Multimodal Transformer is the main contribution of this paper while fusion strategy of multimodal features is missing.
(4) Training details of used networks needed to be described.
(5) The authors should rearrange the structure of this paper to make it more readable.
(6) LSTM、GRU、Conv1D are all traditional methods and there are also many improved transformer networks. The authors should adopt more new methods to compare with their method.
Comments on the Quality of English LanguageThe authors should improve the written English and rearrange the structure of this paper to make it more readable.
Author Response
Dear Reviewer,
We would like to express our gratitude for your valuable feedback and constructive suggestions. We have carefully reviewed your comments and made the necessary revisions to improve the manuscript. These revisions were implemented to the best extent possible, in alignment with the feedback provided by the other reviewers, ensuring consistency across the responses. Below is a summary of the changes we have implemented in response to your comments:
-
Coordinate Fonts in Figures
We appreciate your feedback regarding the small font sizes in the figures. We have revised all figures by increasing the sizes of the coordinate fonts, legends, and labels to ensure that they are now clearly legible. These adjustments improve the readability of the figures both in printed and digital formats. -
Transformer Network Structure and Parameters
Thank you for pointing out the lack of details regarding the Transformer network structure. In response, we have included a comprehensive table outlining the specific architectures and hyperparameters for each model, including the Transformer. The table specifies the number of layers, attention heads, hidden units, dropout rates, learning rates, and other relevant parameters. Additionally, we have expanded the discussion in the methodology section to clarify how these configurations were optimized for the radar signal dataset. -
Fusion Strategy of Multimodal Features
We agree that the fusion strategy is a critical aspect of our approach. To address this, we have clarified the fusion process in the revised manuscript. In the Results section, we provide a detailed explanation of how features (amplitude, phase, kurtosis, and skewness) were extracted from the radar signals and combined into a unified input representation for the Transformer. We describe the preprocessing steps, normalization, and concatenation strategy used to ensure effective integration of the multimodal features. -
Training Details of Networks
We thank you for suggesting the inclusion of training details. We have now added a dedicated subsection in the Results section that provides a comprehensive description of the training process for each network. This includes batch size, number of epochs, optimizer used (e.g., Adam), learning rate schedule, early stopping criteria, and hardware specifications (GPU details). We also discuss the training times and convergence behavior observed during the experiments to provide a clearer understanding of the training setup. -
Structure of the Paper
We appreciate your suggestion to improve the readability and structure of the paper. Our aim was to present a logical progression of the research, starting with the introduction, followed by a detailed description of the dataset and methodology, then the benchmark results with traditional models (LSTM, GRU, Conv1D), and finally the introduction of the multimodal Transformer approach and its results. We acknowledge that this structure may appear extensive, but we have made efforts to ensure that it is coherent and that each section contributes effectively to the overall narrative. We remain open to further restructuring suggestions if you believe additional adjustments are needed. -
Inclusion of Newer Methods for Comparison
We selected LSTM, GRU, and Conv1D as baseline methods because they represent key milestones in the evolution of neural networks for sequential data analysis. LSTM is a foundational method for handling long-term dependencies, while GRU offers a simpler, more computationally efficient alternative. We believe that comparing these traditional methods with our multimodal Transformer approach provides a solid foundation for demonstrating its effectiveness. However, we acknowledge the importance of incorporating newer methods, and we will consider exploring additional recent approaches in future work. -
Note on Manuscript Revision
We would like to inform the reviewers that the manuscript has undergone a comprehensive revision, including thorough language editing and formatting, through the professional service provided by Editage.com. This process ensured that the manuscript meets the highest standards of clarity, coherence, and readability. We believe these revisions have further enhanced the quality of the paper, and we sincerely appreciate the support from Editage.com in refining the manuscript.
We trust that these revisions have addressed your concerns and significantly improved the quality of the manuscript. Thank you once again for your insightful suggestions.
Sincerely,
The Authors
Round 2
Reviewer 2 Report
Comments and Suggestions for Authors
It's a pleasure to accept the paper.